# Anti-CD99 Antibody Therapy Triggers Macrophage-Dependent Ewing Cell Death In Vitro and Myeloid Cell Recruitment In Vivo

**DOI:** 10.3390/antib13010024

**Published:** 2024-03-18

**Authors:** Allison F. O’Neill, Evelyn M. Nguyen, Evelyn D. Maldonado, Matthew R. Chang, Jiusong Sun, Quan Zhu, Wayne A. Marasco

**Affiliations:** 1Department of Pediatric Oncology, Harvard Medical School, Dana-Farber and Boston Children’s Cancer and Blood Disorders Center, Boston, MA 02215, USA; 2Department of Pediatric Oncology, Harvard Medical School, Dana-Farber Cancer Institute, Boston, MA 02215, USA; nguyenm4@sloankettering.edu (E.M.N.); evelyn.maldonado77@gmail.com (E.D.M.); 3Department of Cancer Immunology and Virology, Harvard Medical School, Dana-Farber Cancer Institute, Boston, MA 02215, USA

**Keywords:** Ewing, CD99, antibody, myeloid, PILRa

## Abstract

Background: Ewing sarcoma is a rare tumor of the bone or soft tissues characterized by diffuse membranous staining for CD99. As this tumor remains incurable in the metastatic, relapsed, and refractory settings, we explored the downstream immune implications of targeting CD99. Methods: We discovered a human anti-CD99 antibody (NOA2) by phagemid panning and investigated NOA2 immune cell-mediated cytotoxicity in vitro and in vivo focusing on the myeloid cell compartment, given that M2 macrophages are present in human tumors and associated with a poor prognosis. Results: NOA2 is capable of inducing immune effector cell-mediated Ewing death in vitro via engagement of macrophages. Mice with metastatic Ewing tumors, treated with NOA2, experience tumor growth arrest and an associated increase in intratumoral macrophages. Further, incubation of macrophages and Ewing cells with NOA2, in conjunction with anti-PILRα antibody blockade in vitro, results in the reactivation of previously dormant macrophages possibly due to interrupted binding of Ewing CD99 to macrophage PILRα. Conclusions: These studies are the first to demonstrate the role of human immune effector cells in anti-CD99-mediated Ewing tumor death. We propose that the engagement of CD99 by NOA2 results in the recruitment of intratumoral macrophages. In addition, interruption of the CD99:PILRα checkpoint axis may be a relevant therapeutic approach to activate tumor-associated macrophages.

## 1. Introduction

Ewing sarcoma is a rare tumor of the bone or soft tissues affecting approximately 250 children, adolescents, and young adults each year in the United States [1]. Patients with chemotherapy-responsive localized disease have an excellent 5-year overall survival of greater than 75% [2]. Conversely, the presence of metastatic disease confers a very poor prognosis; less than 20% of these patients are cured [3,4]. The mainstay of therapy for Ewing sarcoma consists of systemic chemotherapy, surgery and/or radiotherapy. While effective for the cohort of patients with localized disease, these treatment modalities confer the life-long risk of organ toxicity, infertility, secondary malignancy, or disfigurement impacting quality of life [5,6,7]. For the subset of patients with metastatic, recurrent, or chemotherapy-refractory disease, overall survival is dismal and little progress has been made in decades [8].

Ewing sarcoma is characterized histologically by small, round blue cells and a diffuse membranous immunohistochemical stain for CD99 (or MIC2), a 32 kDa type I membrane glycoprotein [9,10,11,12,13]. Eighty-five percent or more of Ewing sarcoma cases harbor a chimeric fusion between the RNA binding protein EWS and an ETS family transcription factor (most commonly EWS-FLI1), creating a fusion oncogene (t(11;22)(q24;q12)) deemed relevant to tumor pathogenesis [14,15]. While the published literature suggests an important link between CD99 and EWS-FLI1, this relationship is incompletely characterized [16,17]. In addition, the Ewing immune landscape has traditionally been described as a “cold” immune state. However, the presence of CD68+ tumor-associated macrophages (TAMs) has been linked to poor clinical outcomes [18,19,20].

In the absence of available EWS-FLI1 targeted agents, therapeutic antibodies targeting CD99 have been investigated. Anti-CD99 antibody blockade has been shown to induce Ewing cell aggregation and caspase-independent apoptosis [11], trigger Ewing cell death in p53 wild-type patient-derived cell lines [21], and more recently incite cell death by a process termed methuosis, a form of non-apoptotic cell death characterized by displacement of the cytosol by vacuoles derived from macropinosomes, involving the IGF-1R/Ras/Rac1 signaling pathway [22]. However, the role of the human immune system in anti-CD99-mediated Ewing cell death has not yet been explored.

In this study, we discovered a high-affinity human anti-CD99 antibody, NOA2, capable of binding Ewing sarcoma cells, and subsequently explored the in vitro and in vivo activity of this antibody to engage macrophages, impact tumor growth, and modulate macrophage function. NOA2 can induce Ewing cell death through the engagement of macrophages and antibody-mediated cellular phagocytosis in vitro. Tumors from humanized xenograft mice treated with NOA2 undergo growth arrest and contain intratumoral infiltrates of human macrophages. Furthermore, inhibition of the binding of Ewing CD99 to macrophage PILRα, an inhibitory receptor and CD99 ligand, by dual anti-CD99 and anti-PILRα blockade leads to macrophage reactivation. Single blockade of either target does not achieve this endpoint. Our results suggest that the binding of CD99 by NOA2 and disruption of the CD99:PILRα may modulate myeloid cell activity.

## 2. Materials and Methods

Details surrounding phagemid panning for the identification of anti-CD99 scFv’s, molecular cloning, expression of anti-CD99 monoclonal antibodies, and kinetic binding studies can be found in the Appendix A.

### 2.1. Cell Culture

CD99-positive human Ewing sarcoma cell lines (A673, TC32, SKNEP-1, TTC466, RDES, TC71 and CADO-ES1) were obtained from the laboratories of Drs. Andrew L. Kung (Memorial Sloan Kettering Cancer Center (MSKCC), formerly of Dana-Farber Cancer Institute (DFCI)) and Kimberly Stegmaier (DFCI) where they were previously authenticated and tested negative for mycoplasma. CD99-negative Kelly neuroblastoma cells were obtained from the laboratory of Dr. Rani George (DFCI). A673, SKNEP-1, and Kelly cells were cultured in DMEM (Gibco, Thermo Fisher Scientific Waltham, MA, USA) and TC32, TTC466, RDES, TC71, and CADO-ES1 were cultured in RPMI (Gibco, Thermo Fisher Scientific Waltham, MA, USA), all supplemented with 10% fetal bovine serum (FBS, Gibco, Thermo Fisher Scientific Waltham, MA, USA) and 1% penicillin-streptomycin (Cellgro, Lincoln, NE, USA). Cells were incubated at 37 °C with 5% CO_2_. 293F cells were purchased from Thermo Fisher (Waltham, MA, USA) and cultured in Freestyle medium (Gibco, Thermo Fisher Scientific Waltham, MA, USA). Cells were passaged a maximum of 8–10 times over 4–8 weeks.

### 2.2. Characterization of Anti-CD99 Binding Activity

Ewing sarcoma cells of various lines were incubated in a florescence-activated cell sorting (FACS) tube with 2 µg/mL of each anti-CD99 monoclonal antibody for one hour at 4 °C followed by staining with a FITC-conjugated mouse anti-human IgG (BD Pharmingen, Franklin Lakes, NJ, USA). Titration analyses were subsequently performed with NOA2 given that this candidate antibody demonstrated the strongest binding across Ewing lines. CD99-negative Kelly neuroblastoma cells were used as a negative control. Kinetic interactions between anti-CD99 antibody clones and soluble CD99-Fc (G&P Biosciences, Santa Clara, CA, USA) were measured utilizing the Octet RED system (ForteBio, Fremont, CA, USA). NOA2 and NOA3 were digested to form F(ab’)2 fragments such that the soluble CD99-Fc could be immobilized on human Fc probes as the ligand, while the F(ab’)2 fragments, at concentrates ranging from 0 nM to 100 nM, could serve as the analyte. ForteBio Octet software (Version 10) was utilized to analyze the data. Once binding curves were established, the experiment was repeated using analyte concentrations in a narrow range (10–15 nm for NOA2 and 10–25 nm for NOA3) so as to more accurately calculate each antibody K_D_. The following settings were utilized: baseline 60 s at 1000 rpm, loading for 300 s at 1000 rpm, association for 300 s at 1000 rpm, and dissociation for 600 s at 1000 rpm.

### 2.3. Human Effector Cell Assays

Antibody-dependent cell-mediated cytotoxicity (ADCC) assays were performed with the use of an ADCC Reporter Bioassay Core Kit (Promega, Maidson, WI, USA #G7010, G7018). Serial dilutions of NOA2 or an isotype control were plated with Ewing cells, or CD99-negative Kelly neuroblastoma cells, at concentrations ranging from 0 µg/mL to 2 µg/mL. Effector cells were plated at an effector-to-target cell ratio of 2.5:1 and incubated for 6 h at 37 °C with 5% CO_2_. A Bio-Glo Luciferase Assay was utilized to quantitate cell death. Plates were read on a POLARStar Omega plate reader and relative light units (RLU) were analyzed for each condition after subtracting for background. ADCC was performed in duplicate for each cell line. Error bars denote standard deviation.

Antibody-dependent cellular phagocytosis (ADCP) assays were performed by first isolating human monocytes from human peripheral blood mononuclear cells (PBMCs). PBMCs were plated in MDM medium (DMEM with 10% heat-inactivated human serum, 1% penicillin/streptomycin, 1% L-glutamine, and 50 ng/mL GM-CSF) in a Petri dish for 3 days at 37 °C with 5% CO_2_. At the end of three days, the supernatant was removed and the MDM media was replenished to further culture the adherent macrophage precursors. On day 7, macrophages were incubated with 0.5 mM EDTA in PBS, lifted, and added at a ratio of 1:4 (target: effector cell) to Ewing sarcoma cells stained with PKH26 fluorescent membrane dye (Sigma Aldrich, St. Louis, MO, USA). Isotype IgG or NOA2 was added to the wells and incubated at 37 °C for 4 h. After co-culture, cells were imaged using a Celigo imaging cytometer (Nexcelom Biosciences, Waltham, MA, USA) and then lifted and transferred to a 96-well V-bottom plate for FACS. Cells were then counterstained with FITC-anti-CD14 for 30 min on ice. Engulfed target cells were defined as PKH26+/CD14+. ADCP was performed in duplicate for each cell line. Statistics were calculated using a two-tailed students *t*-test with a *p* value < 0.05 denoting significance.

### 2.4. In Vivo Cytotoxicity Assays

An in vivo experiment was first performed utilizing male and female NOD scid gamma (NSG) mice injected via tail vein with 20,000 TC32 Ewing sarcoma cells transduced with a Luc-mCh vector to allow for bioluminescence imaging (BLI). Mice were approximately 4–6 weeks in age and weighed approximately 25 g. BLI was performed weekly (Xenogen IVIS, Hopkinton, MA, USA) until tumor burden (liver micro-metastases) could be reliably quantitated after 5 weeks of growth. A total of 1 × 10^7^ human peripheral blood mononuclear cells (PBMCs) were injected via tail vein at 5 weeks and mice were treated twice weekly with 5 mg/kg of isotype control (*n* = 2 mice) or NOA2 (*n* = 3) with the first treatment initiated 24 h following PBMC infusion. BLI was performed weekly and regions of interest (ROIs) were drawn over the liver to quantify serial mean luminescence (in units of total flux (p/s)). Mice were euthanized after 4 treatments, prior to the development of graft versus host disease, and tumor tissue harvested for hematoxylin and eosin staining and immunohistochemistry. Immunohistochemistry slides were imaged on a confocal microscope using either a Nikon camera or the Keyence BZ-X800 system (Keyence Co., Itasca, IL, USA). Cells of various immune subtypes were counted from three 10× high-powered fields in representative tumors from mice treated with either IgG or NOA2 and a student’s two-tailed *t*-test (with significance denoted by a *p*-value < 0.05) was run to compare counts. Error bars are included to denote standard deviation. All animal studies were executed in compliance with institutional guidelines and after approval from the Dana-Farber Cancer Institutional Animal Care and Use Committees (IACUC) (DFCI protocol 17-002).

### 2.5. Flow Cytometry of CD45+ Cells Isolated from Subcutaneous Ewing Tumors

A follow-up in vivo experiment, with endpoint analyses to be reported elsewhere, established subcutaneous Ewing sarcoma tumors in humanized mice. Neonatal NSG-SGM3 mice underwent whole-body irradiation (100 cGy) and tail vein injection of 2.5 × 10^5^ CD34+ hematopoietic stem cells (HSCs). After 10–12 weeks, flow cytometric analysis of circulating blood was performed to confirm engraftment defined as >20% peripheral blood human CD45+ cells. Subcutaneous flank tumors were established in 30 mice after injection of 1 × 10^6^ luc-mCh-transduced TC32 cells. Animals were randomized to treatment with 3 mg/kg of isotype IgG (*n* = 12) versus NOA2 (*n* = 15) for a total of 6 treatments delivered over three weeks; treatment began when tumors measured approximately 150–200 mm^3^. Disease remained non-metastatic and confined to the subcutaneous region. Two mice at baseline and one mouse from each treatment cohort were sacrificed after 2, 4 and 5 treatments. Tumors from these timepoints were dissociated to a single-cell suspension, treated with Ficoll to isolate the buffy coat, and prepared for flow cytometry. Cells were gated for human CD45+ expression and the percentage of human CD33+ cells, as a subset of CD45+ cells, was reported. For statistical analysis, tumors from each treatment cohort (*n* = 3 treated with IgG, *n* = 3 treated with NOA2) were grouped and a student’s two-tailed *t*-test was performed with significance denoted by a *p*-value < 0.05. Error bars are included to denote standard deviation. All animal studies were executed in compliance with institutional guidelines and after approval from the Dana-Farber Cancer Institutional Animal Care and Use Committees (IACUC) (DFCI protocol 17-008).

### 2.6. IGF-1 as a Macrophage Chemoattractant

Peripheral blood monocytes (PMNs) were isolated from human PBMCs by double gradient centrifugation [23]. Following isolation, monocytes were diluted to a concentration of 3 × 10^6^ cells/mL in complete RPMI-1640 (Thermo Fisher, Waltham, MA, USA), supplemented with 10% FBS, 100 U/mL Penicillin/Streptomycin, 4.5 g/L D-glucose, 2.383 g/L HEPES, L-Glutamine, 15 g/L sodium bicarbonate, and 110 mg/L sodium pyruvate. A total of 3 × 10^4^ cells were inserted in the upper chamber of a 96-well TC-treated HTS transwell plate (Corning, Corning, NY, USA, CLS3388) with a 5.0 µm pore polycarbonate membrane. A chemotactic gradient was established using different concentrations of MCP-1 (monocyte chemoattractant protein-1) or IGF-1 (0.5 to 100 ng/mL) in media supplemented with 8 µM Hoechst dye (Thermo Fisher, Waltham, MA, USA). After 4 h, the inserts were removed. Cells were spun down and the total fluorescent count from each well, as a reflection of cell migration, was recorded using a Celigo Imaging Cytometer (Nexcelom Bioscience, LLC, Waltham, MA, USA). These experiments were performed in triplicate and statistical significance was calculated using a paired two-tailed student’s *t*-test with significance denoted by a *p*-value < 0.05. Error bars are included to denote standard deviation.

### 2.7. PILRα Synthesis and Analysis of Binding to Human CD99

Human His-tagged PILRα was synthesized by cloning the extracellular domain into a pcDNA3.4 vector and transfecting Expi293F cells (Life Technologies, Foster City, CA, USA). Protein was isolated from the supernatant and purified utilizing a HisPur Ni-NTA column (Qiagen, Venlo, The Netherlands). An SDS PAGE gel demonstrated a band at 55 kDa. Protein synthesis was subsequently confirmed by ELISA, coating a plate with 2 µg/mL of our synthesized PILRα protein and incubating with a commercially available anti-human PILRα antibody (Sigma Aldrich, St. Louis, MO, USA) at a concentration of 2 µg/mL to assure binding. After verification, binding of human PILRα to human CD99 was assessed by ELISA. An ELISA plate was coated with 2 µg/mL of human CD99 and incubated with PILRα at concentrations ranging from 0.25 to 5 µg/mL and an anti-PILRα antibody with HRP-bound secondary.

### 2.8. Macrophage Function in the Context of PILRα and CD99 Blockade

Monocyte-derived macrophages were isolated from PBMCs and identified as the cells left adherent following 5–7 days of PBMC culture. They were primed with 50 ng/mL IFNγ (Novus Biologicals, Centennial, CO, USA) and triggered for M1 polarization with 10 ng/mL lipopolysaccharide (LPS, Sigma Aldrich, Waltham, MA, USA) [24]. Twenty-four hours after LPS treatment, control group macrophages were incubated with one of three antibody conditions: anti-CD99 alone, anti-PILRα alone, or the combination of both antibodies. The experimental group macrophages were co-incubated with either CD99+ TC32 Ewing cells at a ratio of 2:1 (macrophage to Ewing cells) or Kelly CD99-neuroblastoma cells, and subsequently treated with each of the three antibody conditions. This experiment was performed with Ewing cells on two separate occasions using different macrophage human donors and in duplicate, using a third donor, with Kelly neuroblastoma cells. Twelve hours later, the supernatant was collected. ELISA was performed on the supernatant to assess changes in the concentration of TNF-α as a marker of macrophage activation (Abcam, Waltham, MA, USA).

## 3. Results

### 3.1. Characterization of Anti-CD99 Antibodies by Flow Cytometry and Bio-Layer Interferometry

The binding of anti-CD99 mAbs to CD99 was first examined by FACS. Three anti-CD99 antibodies (NOA1, 2, and 3) were isolated by phage display and characterized (Appendix A and Appendix A). FACS binding of IgG1 isoforms to CD99-positive patient-derived Ewing sarcoma cell lines was performed (Figure 1). NOA1 (blue) demonstrated limited binding at concentrations as high as 10 µg/mL, whereas NOA2 (red) and 3 (green) demonstrated strong binding at concentrations of 2 µg/mL (Figure 1A (left)). None of the clones bound to the CD99-negative Kelly neuroblastoma cells (Figure 1A (right)). NOA2 was later used for further studies given superior binding across Ewing cell lines (results not shown). The relative affinity of NOA2 was determined by flow cytometric saturation binding studies against five independent CD99-positive Ewing sarcoma cell lines (Figure 1B; Appendix A demonstrates replicates for A673, TC71, and TTC446). NOA2 binding was specific for CD99-positive Ewing cells and demonstrated circa 3-fold range in binding affinity across five Ewing lines. OctetRed was performed and revealed a Kd 3.814 × 10^−10^ M ± 5.827 × 10^−12^ M with K_on_ 2.068 × 10^5^ ± 1.065 × 10^3^ (1/Ms) and K_off_ of 7.887 × 10^−5^ ± 1.135 × 10^−6^ (1/s) (Figure 1C). Figure 1D demonstrates the heavy and light chain sequences and comparative human germlines for the NOA2 scFv as identified by IMGT [25,26].

### 3.2. Anti-CD99 Antibodies Mediate Killing of CD99+ Ewing Sarcoma Cells by ADCC and ADCP

To determine if immune effector cells play a role in NOA2 antibody-mediated cytotoxicity, ADCC and ADCP were performed. NOA2 triggered Ewing cell death as a function of antibody concentration in ADCC assays; however, the magnitude of cell killing was cell-line specific. As shown in Figure 2A–C, Ewing cell death as assessed by net luminescence was more pronounced for CADO-ES1 and TC32. Ewing cells retained viability when incubated with an isotype control and CD99-negative Kelly neuroblastoma cells.

For in vitro ADCP studies, macrophages isolated from human PBMCs and co-incubated with Ewing sarcoma cells and NOA2 showed aggregation and engulfment of tumor cells (Figure 2D, arrows). Similar to ADCC, these ADCP results were more pronounced for CADO-ES1 and TC32. FACS demonstrated an increase in PKH26+/CD14+ cells, representing Ewing cells engulfed by macrophages following NOA2 incubation; the two NOA2 concentrations tested for ADCP (2 and 5 µg/mL) did not show an antibody dose-dependent effect (Figure 2E,F). ADCP was most prominent visually following incubation with 2 µg/mL, suggesting that lower concentrations of NOA2 may be more effective for this immune effector function. The lack of a dose-dependent effect on the percent of FITC+/PE+ cells on flow cytometry may be secondary to an increase in cell death or clumping at the higher antibody concentration. There was comparatively minimal aggregation and engulfment of Ewing sarcoma cells with the incubation of isotype IgG. In summary, NOA2 kills Ewing cells by both ADCC and ADCP, with differing efficiency across cell lines, implicating immune cells in antibody-mediated cytotoxicity.

### 3.3. In Vivo Ewing Tumor Growth Arrest Following Treatment with NOA2 and Resulting Myeloid Cell Infiltrate

We next sought to determine whether anti-CD99 antibody treatment of mice harboring micro-metastatic Ewing sarcoma tumors could inhibit tumor growth. An in vivo experiment was performed with a human peripheral blood leukocyte (PBL)-NSG humanized mouse model harboring Ewing sarcoma liver micro-metastases. Mice were treated with NOA2 (*n* = 3) or an isotype control (*n* = 2), initiated 24 h after PBMC injection, for a total of four treatments (arrows). Figure 3A demonstrates tumor growth arrest at 14 days post-treatment, as extrapolated by mean luminescence, in mice treated with NOA2 as compared with an isotype control.

To investigate NOA2’s mode of action, mice were euthanized following therapy and livers dissected, embedded in paraffin, and sectioned (Figure 3B). The representative images demonstrate a more prominent human CD45, mouse CD14, and human CD33 and CD16 infiltrate. There was a statistically significant increase in human CD45+, mouse CD14+, and human CD33+ infiltrating cells (*p* < 0.05) in mice treated with NOA2 compared to IgG when averaged over three 10× high-powered fields. The differences for human CD16+ cells (*p* < 0.16) did not reach statistical significance, possibly due to small tumors and low overall cell counts (Figure 3C). In NOA2 treated mice, there was also a noticeable difference in the appearance of mouse myeloid cells staining for CD14; these cells were more plump and more aggregated than in IgG-treated tumors, suggesting macrophage activation, similar to that seen in in vitro ADCP studies. Anti-CD99 treated tumors also demonstrated pockets of necrosis indicative of apoptosis by TUNEL staining (Figure 3B).

The observation that treatment with NOA2 led to an accumulation of hCD45+ and hCD33+ immune cells in Ewing tumors in the PBL-NSG mouse study above led us to perform a second treatment study in CD34+ HSC-reconstituted NSG-SGM3 mice which show enhanced engraftment of human myeloid cells [27,28]. Mice were treated twice weekly with either IgG or NOA2 and subcutaneous tumors were harvested from mice at baseline and from each cohort following treatments 2, 4, and 5. Human CD45+ cells were isolated from the harvested subcutaneous tumors and analyzed by FACS. Cells were gated for human CD45+ expression; the percentage of CD45+ cells expressing human CD33+ at baseline compared to the three timepoints is shown in Figure 3D. Tumors from the isotype or NOA2 treatment cohorts were grouped to allow for statistical analyses. As shown in Figure 3E, there is a robust human CD33+ myeloid cell infiltrate in NOA2-treated mice with ~55% of hCD45+ cells being derived from the myeloid lineage (*p* < 0.01). In summary, these collective results demonstrate that NOA2 treatment of humanized mice bearing Ewing tumors leads to the recruitment of CD33+ monocytes/myeloid cells.

### 3.4. IGF-1 Mediated Macrophage Chemotaxis

Given prior reports documenting upregulation of the IGF/IGF-1R axis with Ras/Rac1 signaling in Ewing sarcoma, we performed immunohistochemistry on tumors from the micro-metastatic in vivo experiment. Figure 4A demonstrates more prominent staining for IGF-1R, Ras, and IGF-1 in tumors from mice treated with NOA2 as compared with IgG. We sought to determine whether IGF-1 was contributing to recruitment of blood monocytes to the tumors. Monocyte trafficking was analyzed in a transwell assay utilizing a chemotactic gradient of either MCP-1 (positive control, Figure 4B left) or IGF-1 (Figure 4B right). A statistically significant increase in monocyte chemotaxis, compared with control, was observed with IGF-1 at doses greater than 10 ng/mL.

### 3.5. Macrophage Activation Following Co-Incubation of Ewing Sarcoma Cells with Anti-CD99 and Anti-PILRα Antibodies

PILRα is a CD99 ligand, and inhibitory receptor, that is preferentially expressed on human myeloid cells but also on natural killer cells, dendritic cells, and other granulocytes. PILRα has been implicated in and can be restrictive of immune cell recruitment and activation [29,30,31,32]. Mouse PILRα has been shown to bind mouse CD99 but this protein interaction has not been well-defined in humans [33]. As shown in Figure 5A, we confirmed dose-dependent binding of human PILRα to soluble human CD99 by ELISA.

With the knowledge that human macrophages express PILRα [24] and confirmation that human PILRα binds human CD99, we next sought to evaluate the effects of blocking the macrophage PILRα:CD99 Ewing sarcoma binding axis (Figure 5B, left panel) on macrophage activity using TNF-α secretion as a proxy. In addition to expressing PILRα, macrophages, like Ewing cells, also express CD99 which can be bound by NOA2. As shown in Figure 5C (upper and lower panels) for all three donors, TNF-α secretion decreases when M1 macrophages (dark bars) are incubated with the anti-CD99/anti-PILRα cocktail, but not with each antibody alone, suggesting that an inhibitory axis is triggered by the binding of both macrophage surface proteins (Figure 5B, right upper panel). This effect persists when M1 macrophages are incubated with CD99-negative Kelly cells (Figure 5C, lower). However, when the M1 macrophages are incubated with CD99-positive TC32 cells, interrupting the binding of the PILRα:CD99 axis (Figure 5B, right upper panels), TNF-α secretion is restored perhaps due to the saturation of anti-CD99 on Ewing cells diverting binding from M1 macrophages. Further mechanistic studies will be required to determine this mode of action and analyze the anti-tumor outcome of this combination immunotherapy in vivo.

## 4. Discussion

Here we report on the discovery of NOA2, a human monoclonal antibody that targets CD99 and is capable of recruiting and reactivating components of the innate immune system to direct antibody-mediated Ewing sarcoma death. NOA2 can trigger antibody-mediated tumor death through ADCC (FcγR-mediated killing) and ADCP (phagocytosis). In vivo treatment of Ewing tumor-bearing mice with NOA2 results in a tumoral infiltrate of morphologically activated mouse CD14+ cells and an increase in human CD33+ cells. NOA2 is also associated with upregulation of IGF-1, potentially contributing to monocyte/macrophage chemotaxis and recruitment. Combination NOA2 and PILRα blockade in vitro potentially reverses the inhibitory CD99:PILRα pathway linking Ewing sarcoma cells and macrophages, leading to restoration of TNF-α secretion. In light of published data showing M2 tumor-protective macrophages to be the predominant infiltrating immune subtype in human Ewing tumors, therapeutic mechanisms aimed at macrophage recruitment and activation are worthy of further study [19,20]. Our studies focused predominantly on M1 macrophages, given recent data suggesting that ligation of CD99 and use of agents targeting the PI3/AKT pathway can promote an M1 phenotype [34,35].

We investigated several immune-mediated mechanisms of action to further understand the tumoricidal activities of NOA2. We observed the differential ability of NOA2 to induce ADCC and ADCP in CADO-ES1 and TC32 cells as compared with A673 cells. A673 cells harbor a BRAFV600E mutation, which is unusual in Ewing sarcoma, and may have an atypical driver role besides the EWS fusion gene [36]. We attempted to reconcile the differences in ADCP killing by examining the potential overexpression of CD47, recognizing that the CD47/SIRPα axis has been implicated in the control of myeloid cell activation [37]. At baseline, human Ewing-derived cell lines express CD47; however, we could not demonstrate a clear upregulation of CD47 on A673 cells following treatment with NOA2 at concentrations of 1 µg/mL or higher (Appendix A).

CD99 antibody blockade has been shown to trigger Ewing cell death through upregulation of the IGF-1R/Ras/Rac1 signaling pathway and a process termed methuosis [22]. While our data corroborate the upregulation of this pathway following binding of NOA2 therapy, NOA2 does not trigger Ewing cell death in the absence of immune cell engagement. This prompted us to evaluate whether the IGF-1R/Ras/Rac1 signaling pathway serves a different purpose following NOA2 binding. The EWS-FLI1 fusion protein has been shown to activate the IGF1 promotor and induce IGF-1 expression [38]. We hypothesize that NOA2 may serve as an agonist on Ewing cells, upregulating EWS-FLI1, leading to enhanced IGF-1 transcription (Appendix A). We have shown by immunohistochemistry that IGF-1 is upregulated in the tumors of NOA2-treated mice and that this protein can serve as a monocyte chemoattractant (Figure 4), possibly explaining the more prominent myeloid infiltrate visualized in NOA2- vs. IgG-treated tumors from our in vivo studies. Mouse CD14+ cells from our initial in vivo experiment appeared morphologically activated, similar to those seen in our ADCP studies. On the basis of these findings, we postulate that NOA2 is responsible for recruiting activated intratumoral macrophages which may contribute to early tumor growth arrest and apoptosis, as visualized by TUNEL staining [39].

Our exploration of the PILRα:CD99 axis was an attempt to address whether macrophages can be re-activated through the disruption of PILRα:CD99 binding. Macrophages are known to express both surface PILRα and CD99, while Ewing cells express only CD99; our NOA2 antibody binds both macrophage CD99 (Appendix A) and Ewing CD99, while the mouse anti-human PILRα antibody we utilized only binds macrophages. We show that human PILRα binds to human CD99, thereby suggesting a mechanism by which human macrophages and Ewing cells might interact [33]. When macrophages are incubated with both anti-CD99 and anti-PILRα antibodies, TNF-α secretion decreases, suggesting that both of these cell surface proteins are implicated in macrophage activity. When macrophages are co-incubated with a CD99-negative Kelly neuroblastoma cell line and incubated with both antibodies, there is little change in TNF-α decline. Incubation of M1 macrophages with Ewing CD99+ cells, conversely, introduces additional CD99 binding sites as well as a novel Ewing—macrophage CD99: PILRα axis predicted to contribute to macrophage suppression. In this context, incubation with both anti-CD99 and anti- PILRα antibodies interrupts the Ewing—macrophage axis and likely also diverts NOA2 binding to Ewing CD99—the combination of which reverses macrophage suppression, allowing a rebound in TNF-α secretion. These experiments are limited by the number of macrophages isolated from each donor, precluding multiple experiments, and donor variability in macrophage function. These early data support additional exploration of the CD99: PILRα axis.

## 5. Conclusions

The study of tumor-infiltrating lymphocytes has received considerable attention, particularly given the striking successes in the use of PD-1 inhibitors for adult malignancies. However, these agents are less likely to be of benefit in Ewing sarcoma given that Ewing tumors are mutationally silent and harbor a paucity of T-cells [18,19,20,40]. This is the first study to demonstrate the role of myeloid cells in antibody-mediated Ewing cell death and to identify a novel immune axis capable of reactivating tumor-associated macrophages. Should anti-CD99 therapeutics gain further traction for study in patients with Ewing sarcoma, important considerations for on-target, off-tumor toxicity will be relevant given the expression of CD99 on other human tissues, most notably leukocytes and pancreatic beta-islet cells [41,42] (https://www.proteinatlas.org). For the population of Ewing sarcoma patients with metastatic, relapsed, or refractory disease, there is a grave need for novel therapeutics. For patients with single-site disease, cure still comes at the expense of significant long-term toxicities. Pivoting focus towards the manipulation of the myeloid compartment may, therefore, be of therapeutic relevance for Ewing sarcoma and other mutationally silent tumors for which novel therapies are sorely lacking.

## Figures and Tables

**Figure 1 antibodies-13-00024-f001:**
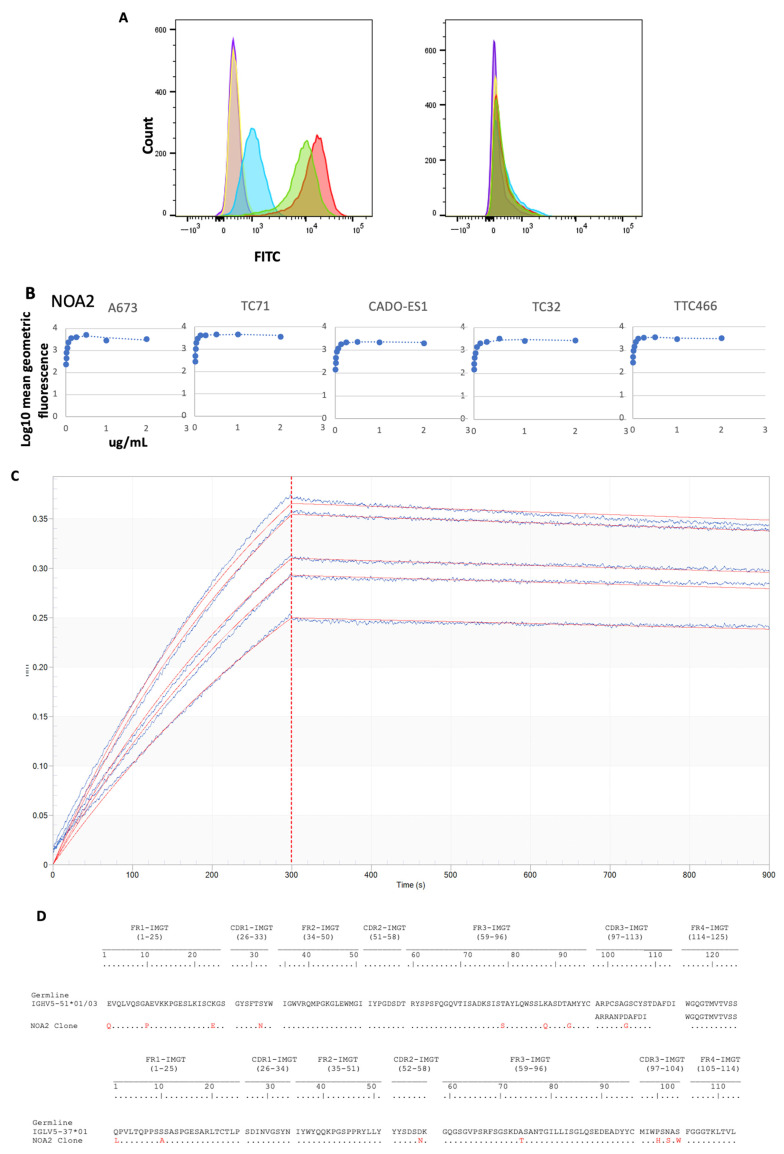
(**A**) (left) FACS for A673 Ewing sarcoma cells alone (purple), A673 cells incubated with a non-specific FITC-bound isotype control (orange), A673 cells incubated with 10 µg/mL of NOA1 (blue), 2 µg/mL of NOA2 (red), and 2 µg/mL of NOA3 (green). All three samples were incubated with a mouse anti-human FITC-conjugated secondary. The right panel depicts FACS for Kelly neuroblastoma alone and in each of the above conditions. (**B**) FACS demonstrating diminishing fluorescence with decreasing concentrations of NOA2 bound to A673, TC32, CADO-ES1, TC71, and TTC466 Ewing sarcoma cells. (**C**) OctetRED analysis for NOA2. NOA2 F(ab’)2 concentrations were as follows: 15 nM (Sensor A), 13.5 nM (Sensor B), 12.5 nM (Sensor C), 11.4 nM (Sensor D), 10 nM (Sensor E), and 0 nM (Sensor F). (**D**) Heavy and light chain sequences, comparative to germline, for NOA2.

**Figure 2 antibodies-13-00024-f002:**
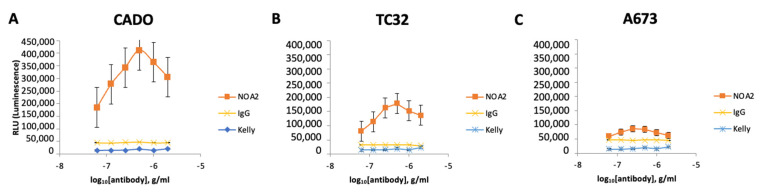
(**A**–**C**) Cell death of three human-derived Ewing sarcoma cell lines (CADO-ES1, TC32, and A673) by ADCC following incubation with anti-CD99 mAb and Jurkat effector cells. Kelly neuroblastoma cells were used as a negative control. (**D**) Light images obtained utilizing Celigo demonstrating aggregated, plump macrophages (arrows) actively engulfing Ewing sarcoma cells following incubation with NOA2 (labeled red with PKH26), i.e., ADCP. This finding was more prominent for the p53-wildtype cell lines CADO-ES1 and TC32 (**A**,**D** arrows). (**E**) A potential dose-independent increase in PKH26+/CD14+ cells (i.e., Ewing cells engulfed by macrophages) following treatment with NOA2 in CADO-ES1 and TC32, but this is not appreciable in the graphical representation of the data (**F**). ns = non-significant.

**Figure 3 antibodies-13-00024-f003:**
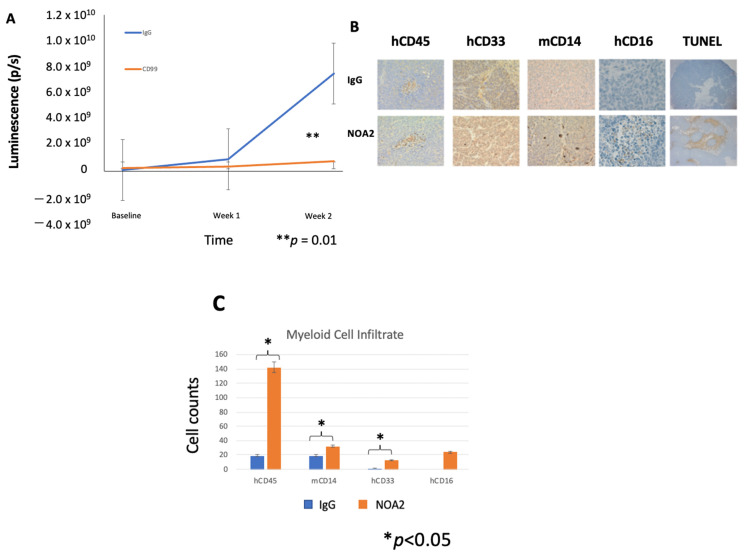
(**A**) Mean fluorescence measured by BLI for mice treated with an isotype control (*n* = 2) versus NOA2 (*n* = 3). Arrows correspond with timed mouse treatments. (**B**) Tumors from mice treated either with IgG or NOA2 embedded in paraffin and stained by immunohistochemistry. Tumors from mice treated with NOA2 demonstrate a more prominent human CD45+ infiltrate, human CD16+/CD33+ staining myeloid cells, and plump activated-appearing mouse CD14+ cells compared with tumors from mice treated with IgG. NOA2-treated tumors also stain for TUNEL indicating apoptosis. (**C**) The numbers of each infiltrating immune cell type counted in three 10× high-powered fields. (**D**) Flow cytometry results of the immune cell infiltrate isolated from subcutaneous tumors in a follow-up humanized mice experiment. Tumors were isolated from one mouse in each treatment cohort at each of the timepoints indicated. Tumors were grouped to allow a calculation of statistical significance between cohorts (**E**).

**Figure 4 antibodies-13-00024-f004:**
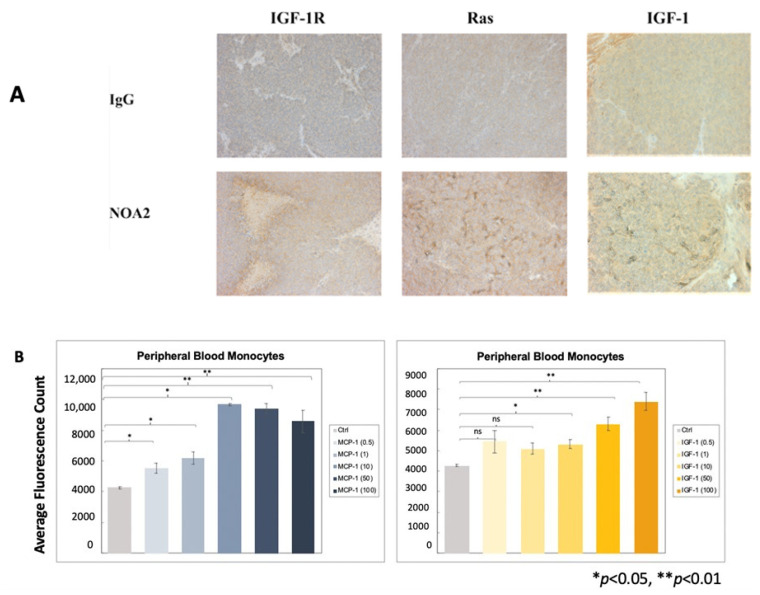
(**A**) Immunohistochemical staining of tumors from the micro-metastatic mouse model depicted in Figure 3A–C. Tumors from mice treated with NOA2, as compared with mice treated with IgG, show upregulation of IGF-1, Ras, and IGF-1, implicating the IGF-1/Ras-Rac-1 pathway downstream of CD99 binding. (**B**) Results to a chemotaxis assay investigating whether IGF-1 is responsible for monocyte migration. Human monocytes migrate through a transwell membrane towards concentrations of IGF-1 greater than 10 ng/mL. The same assay was performed using MCP-1 as a positive control (left). ns = non-significant.

**Figure 5 antibodies-13-00024-f005:**
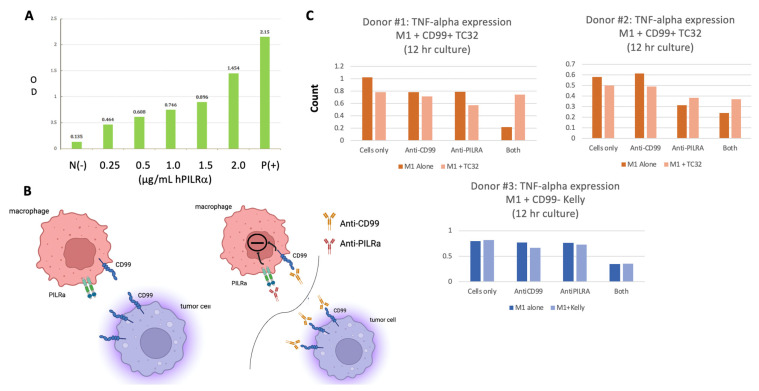
(**A**) ELISA results confirming human PILRα binding to human CD99 in a PILRα-dose dependent fashion. (**B**) The proposed interaction between macrophage PILRα and Ewing CD99 (left) as well as the potential inhibitory pathway triggered by binding both macrophage CD99 and PILRα (right upper). Disruption of the CD99: PILRα macrophage–Ewing checkpoint pathway is also depicted (right lower). (**C**) The rebound in macrophage TNF-α secretion detected following co-incubation of macrophages with Ewing cells and both anti-PILRα and anti-CD99 antibodies.

## Data Availability

The data presented in this study are available on request from the corresponding author.

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
