# Peer review of "Anti-CD99 Antibody Therapy Triggers Macrophage-Dependent Ewing Cell Death In Vitro and Myeloid Cell Recruitment In Vivo"

_2073-4468, 2024, doi:10.3390/antib13010024_

Round 1

Reviewer 1 Report

Comments and Suggestions for Authors

The authors present the isolation and in vitro as well as in vivo characterization of a novel anti-CD99 antibody for treatment of Ewing sarcoma. The manusript is well written and the experiments logically and acurately designed, performed and analysed. They reveal macrophage dependent killing of the tumor cells which can be increased by combination treatment with PILRalpha antibody.

main issue:

all figures are far to small, especially labelling, and on a normal printout the figures cannot be understood and many things cannot be recognized. Please enlarge!

minor comments:

- authors should elaborate more on the expression of CD99, not only on tumor cells, but also on normal cells

- please include in the main text some information on the sequence analysis of the isolated antibody, the measured KD values and eventually information on epitope location.

- in Fig.1 a semi-log presentation of the data would be more helpful.

- in the discussion please explain "methosis".

Reviewer 2 Report

Comments and Suggestions for Authors

This comprehensive study presents a novel avenue for treatment of a rare sarcoma type, which is particularly appealing as the newly discovered NOA2 antibody can efficiently activate the immune response.

The statistics employed needs to be specified in the Methods section and in the legends of the respective figures, e.g. whether SD or SEM are indicated. SD should be preferentially used.

Furthermore, the prospects of exploring similar therapies in other mutationally silent tumors should be concisely indicated.

Reviewer 3 Report

Comments and Suggestions for Authors

This manuscript demonstrates the discovery of a novel anti-CD99 antibody, NOA2, and how this antibody can trigger immune-mediated killing of Ewing sarcoma and recruitment of monocytes in vitro and in vivo. This is a nice story, however some additional experiments need to be performed for this study.

Figure 1:

-       Is there errors/standard deviations for your EC50 calculations? Was this binding n=1? Would like to see some repeats, especially as some of the points for A673 are very varied. 

-       Although the FACS binding is okay, some SPR analysis to determine affinity of NOA2 (and NOA1 and NOA3) would be nice

Figure 2:

-       Do you have expression levels of CD99 for the three cell lines? Does this correspond to the differences in ADCC?

-       Is the killing significant for A673? Again, how many repeats were performed? If just n=1, this is not sufficient for publication. 

-       “ACDP was most prominent visually following 2ug/ml” – the images do look like ADCP is stronger in 2ug/ml but your calculations are almost identical for 5ug/ml? This does not seem consistent. i.e. for TC32, the image at 2ug/ml looks like there are significantly more red cells than 5ug/ml but the bar graph shows pretty much identical. Are there just less cells at 5ug/ml or have some died? Is this just the selection of image? 

-       Use of lower concentrations may be nice to demonstrate dose-dependent effects 

-       In the introduction, you say that Anti-CD99 antibody blockade has previously been shown to induce Ewing cell aggregation and caspase-independent apoptosis. Does your NOA2 do this? Not all antibodies induce the same target effects – i.e. trastuzumab and pertuzumab. Do you know how of your ADCC is Fc-mediated and how much is direct effects? 

Figure 3: 

-       Title written twice in figure 3E

Figure 5: 

-       Can you normalise the results from the donors and combine to get some statistics? One or two donors is not particularly compelling, more repeats are needed

-       Why did you look at M1 macrophages when you acknowledge in the discussion that M2 macrophages are the predominant subtype in Ewing tumours? Is there a difference in CD99 and PILRa expression in M1 vs M2 macrophages? A repeat of this experiment with M2 macrophages would be more relevant to the disease type you are studying. 

Also, in general, the figures are quite small and hard to read - some larger labels would be nice throughout. 

Round 2

Reviewer 3 Report

Comments and Suggestions for Authors

Thank you to the authors for taking on board the comments, the manuscript has improved. However, I have one minor concern to be addressed: 

Figure 2 - the error bars that have been added in seem to be the same error bar across all conditions? The error bar is above the mean point for the first concentration, this does not seem right. Can the authors please fix the error bars so it is the correct error bar for each concentration? 

Author Response

Thank you; apologies for the error in formatting.  I have revised Figure 2 to reflect accurate error bars denoting standard deviation.